# Palliative Quad Shot Radiation Therapy with or without Concurrent Immune Checkpoint Inhibition for Head and Neck Cancer

**DOI:** 10.3390/cancers16051049

**Published:** 2024-03-05

**Authors:** Rituraj Upadhyay, Emile Gogineni, Glenis Tocaj, Sung J. Ma, Marcelo Bonomi, Priyanka Bhateja, David J. Konieczkowski, Sujith Baliga, Darrion L. Mitchell, Sachin R. Jhawar, Simeng Zhu, John C. Grecula, Khaled Dibs, Mauricio E. Gamez, Dukagjin M. Blakaj

**Affiliations:** 1Department of Radiation Oncology, The Ohio State University Wexner Medical Center, Columbus, OH 43210, USA; rituraj.upadhyay@osumc.edu (R.U.); emile.gogineni@osumc.edu (E.G.); glenis.tocaj@osumc.edu (G.T.); sungjun.ma@osumc.edu (S.J.M.); david.konieczkowski@osumc.edu (D.J.K.); sujith.baliga@osumc.edu (S.B.); darrion.mitchell@osumc.edu (D.L.M.); sachin.jhawar@osumc.edu (S.R.J.); simeng.zhu@osumc.edu (S.Z.); john.grecula@osumc.edu (J.C.G.); khaled.dibs2@osumc.edu (K.D.); 2Department of Medical Oncology, The Ohio State University Wexner Medical Center, Columbus, OH 43210, USA; marcelo.bonomi@osumc.edu (M.B.); priyanka.bhateja@osumc.edu (P.B.); 3Department of Radiation Oncology, Mayo Clinic, Rochester, MN 55905, USA; gamezharo.mauricio@mayo.edu

**Keywords:** head and neck cancer, radiation therapy, quad shot, immunotherapy, local control, immune checkpoint inhibition, metastatic, recurrent, palliative, radiotherapy

## Abstract

**Simple Summary:**

Immunotherapy represents the standard-of-care systemic therapy in patients with recurrent and metastatic head and neck cancer (HNC). However, these patients often present with local disease, which can affect quality of life, and local progression can lead to significant morbidity. Local response rates from immunotherapy alone are subpar. Thus, palliative radiation is often utilized in this scenario. ‘QuadShot’(QS), a hypofractionated palliative radiation regimen, can provide symptomatic relief and local control and may potentiate the effects of immunotherapy. There have been no previous studies evaluating the combination of immunotherapy with QS. We found that the combination of QS with concurrent immunotherapy was well tolerated and significantly improved local control compared to QS alone. The median survival of 9.4 months compares favorably to historical controls for patients with HNC treated with QS. This approach represents a promising treatment option for patients with HNC unsuited for curative-intent treatment and warrants prospective evaluation.

**Abstract:**

Objectives: Patients with recurrent and metastatic head and neck cancer (HNC) have limited treatment options. ‘QuadShot’ (QS), a hypofractionated palliative radiotherapy regimen, can provide symptomatic relief and local control and may potentiate the effects of immune checkpoint inhibitors (ICIs). We compared outcomes of QS ± concurrent ICIs in the palliative treatment of HNC. Materials and Methods: We identified patients who received ≥three cycles of QS from 2017 to 2022 and excluded patients without post-treatment clinical evaluation or imaging. Outcomes for patients who received QS alone were compared to those treated with ICI concurrent with QS, defined as receipt of ICI within 4 weeks of QS. Results: Seventy patients were included, of whom 57% received concurrent ICI. Median age was 65.5 years (interquartile range [IQR]: 57.9–77.8), and 50% patients had received prior radiation to a median dose of 66 Gy (IQR: 60–70). Median follow-up was 8.8 months. Local control was significantly higher with concurrent ICIs (12-month: 85% vs. 63%, *p* = 0.038). Distant control (12-month: 56% vs. 63%, *p* = 0.629) and median overall survival (9.0 vs. 10.0 months, *p* = 0.850) were similar between the two groups. On multivariable analysis, concurrent ICI was a significant predictor of local control (HR for local failure: 0.238; 95% CI: 0.073–0.778; *p* = 0.018). Overall, 23% patients experienced grade 3 toxicities, which was similar between the two groups. Conclusions: The combination of QS with concurrent ICIs was well tolerated and significantly improved local control compared to QS alone. The median OS of 9.4 months compares favorably to historical controls for patients with HNC treated with QS. This approach represents a promising treatment option for patients with HNC unsuited for curative-intent treatment and warrants prospective evaluation.

## 1. Introduction

Head and neck cancer (HNC) represents a significant cause of morbidity and mortality worldwide [1]. Approximately 10% of patients with HNC present with metastatic disease at initial diagnosis, with an additional 40–60% of patients progressing after definitive treatment [2]. These patients are often ineligible for curative-intent treatment. Additionally, a small percentage of patients with locally advanced disease may also not qualify for definitive intent treatment due to age, comorbidities, performance status, tumor bulk, and/or previous radiation. Locoregional disease, left untreated, can lead to progressive dysphagia, airway obstruction, pain, aspiration, and bleeding, which can deteriorate quality of life and negatively impact survival. Palliative radiation therapy (RT) is often utilized for these patients to prevent local progression and alleviate symptoms caused by locoregional disease. Several palliative regimens have been studied, including conventional fractionation, hypo- and hyperfractionation, and stereotactic body RT [3,4,5,6,7,8,9,10,11,12,13].

‘Quad Shot’ (QS), a cyclical hypofractionated palliative RT regimen, was first investigated prospectively in Radiation Therapy Oncology Group (RTOG) 8502 for advanced pelvic malignancies. This regimen consists of cycles of 14.8 Gy delivered in four twice-daily fractions (3.7 Gy per fraction) over 2 consecutive days, with cycles given every 3–4 weeks, typically to a total of up to three cycles [14,15]. QS was subsequently studied for HNC, with results showing it has the potential to provide symptomatic relief and contribute to local control without significant toxicity [13,16,17,18,19,20]. However, QS radiation alone may be insufficient for long-term disease control, with median duration of local control and overall survival of less than 6 months [16,21].

The emergence of immunotherapy has led to significant improvement in outcomes for patients with advanced cancers [22,23]. Immune checkpoint inhibitors (ICIs) now represent the standard of care for non-nasopharyngeal recurrent and metastatic HNC, after results from KEYNOTE-048 and CheckMate 141 showed improved overall survival with the use of Pembrolizumab and Nivolumab, respectively [24,25,26]. However, trials investigating the use of ICIs for recurrent and metastatic HNC have suggested suboptimal response rates of <20% for gross disease [24,25,26,27,28,29]. The benefit of ICIs in combination with RT is poorly defined, with recent trials evaluating the addition of ICIs to chemoradiation for locally advanced HNC showing no clear improvement in control or survival [30,31,32].

The combination of QS and ICIs is the subject of ongoing trials (NCT04454489 [33], NCT04373642 [34]). However, there have been no published studies to date to our knowledge evaluating the benefit of the addition of ICIs to QS. We compared response rates, local and distant control, survival, and toxicity in patients with HNC treated with QS with or without concurrent ICIs. We hypothesized that the addition of ICIs to QS would improve response to treatment and correspondingly increase local control in comparison with patients treated with QS alone.

## 2. Materials and Methods

### 2.1. Patient Selection and Study Design

We conducted an Institutional Review Board-approved retrospective analysis of patients with HNC treated with QS from January 2017 to December 2022 at our National Cancer Institute-Designated Cancer Center. Patients with primary cancers of the head and neck region (cutaneous or mucosal) who received at least 3 cycles of QS RT were included for analysis. Patients without follow-up clinical examination or imaging available for review after QS were excluded.

Pre-treatment evaluation included a medical history, physical examination, complete blood cell count (CBC), complete metabolic profile (CMP), and appropriate radiographic studies to assess disease extent. Disease was staged based on the extent of disease upon initial presentation at diagnosis using the American Joint Committee on Cancer (AJCC) Cancer Staging Manual, Seventh Edition. All patients were discussed in a multidisciplinary conference. Patients who were considered for QS included those with recurrent and/or metastatic HNC, in addition to those with locally advanced disease who were ineligible for curative intent treatment due to disease extent or comorbidities.

### 2.2. Radiation Simulation and Treatment Planning

All patients were simulated supine with a head and neck thermoplastic mask prior to each planned QS cycle. Gross tumor volume (GTV) was delineated by the treating radiation oncologist based on clinical examination and available imaging. No margin from GTV was added for clinical target volumes (CTVs) in the setting of re-irradiation, while an optional 3–5 mm margin was allowed for patients who had not undergone previous RT. An additional 3–5 mm isometric expansion from CTV was added to create planning target volumes (PTVs). Daily 5 mm surface bolus was used for patients with skin invasion. No elective neck lymph nodal coverage was included in treatment volumes. Patients were treated on a Varian linear accelerator (Varian, Palo Alto, CA, USA) using 3-dimensional conformal RT (3D-CRT) or volumetric modulated arc therapy (VMAT). Prescription dose to the PTV was 3.7 Gy per fraction given twice daily (BID) separated by an interval of at least 6 h for two consecutive days, for a total dose of 14.8 Gy per cycle. This was repeated every 3–4 weeks as tolerated up to a total of 3–5 cycles. Pre-treatment kV cone beam CT was used for image guidance before each fraction. Patients were examined and simulated prior to each QS cycle to evaluate for response to treatment, with subsequent plan adaptation when significant changes in treatment volume were noted.

### 2.3. Systemic Therapy

Patients received concurrent systemic therapy based on their performance status and comorbidities per multidisciplinary discussion. Administration of ICIs was considered concurrent with QS when delivered within 4 weeks of RT. The most commonly used ICI regimens included Pembrolizumab 200 mg fixed dose every 3 weeks and Nivolumab 480 mg fixed dose every 4 weeks. PD-L1 status was not routinely obtained but was recorded when available.

### 2.4. Follow-Up and Assessments

Patients were followed every 2–3 months with clinical examination and cross-sectional imaging (contrast-enhanced CT neck, MRI neck, and/or PET/CT) after completing QS. The last clinic visit, imaging, or date of contact was used for censoring patients alive at the time of analysis. Follow-up data collected included adverse events, response rates, local and distant progression, and death. Local progression was defined as tumor progression within the RT-treated field (encompassed by 95% isodose line), while any failure outside the RT field, including in untreated lymph nodal groups, was considered distant progression. All endpoints were defined as the interval between completion of cycle 1 QS radiation to the event. Local and distant control were defined as time to date of local and distant progression, respectively. Overall survival (OS) was defined as time to death from any cause. Patients who were lost to follow-up were censored at that timepoint. Post-treatment tumor response was assessed using the modified Response Evaluation Criteria in Solid Tumors criteria (RECIST version 1.1) [35]. Adverse events were graded per Common Terminology Criteria for Adverse Events version 5.0 (CTCAE v5.0) acute and late toxicity grading scales.

### 2.5. Statistical Analysis

Summary statistics for patient characteristics are presented as median and interquartile ranges (IQRs). The Pearson chi-squared test for categorical variables and Wilcoxon rank-sum test for continuous variables were used to assess measures of association in frequency tables. Kaplan–Meier curves were used for survival analyses and the log-rank test was used for intergroup comparisons. Fine and Gray competing risks regression was used to evaluate cumulative incidence of local failure, where death without the outcome was a competing event. Univariate (UVA) and multivariable (MVA) analyses using Cox proportional hazards models were conducted to evaluate the associations between pertinent clinical factors and outcomes. Variables with a *p*-value < 0.200 were included in the MVA model. Both Cox proportional hazards and Fine and Gray competing risks regression analyses were summarized using hazard ratios (HRs) and their 95% confidence intervals (CIs). *p*-values < 0.050 were considered statistically significant. Statistical tests were based on a 2-sided significance level. All statistical analyses were performed using SPSS v23.0 (IBM Corp, Armonk, NY, USA) and R v4.2.2 (R Core Team, R Foundation for Statistical Computing, Vienna, Austria).

## 3. Results

Figure 1 outlines the patient selection schema.

A total of 70 patients were included for analysis with a median follow-up of 8.8 months (range: 3.2–75.3, IQR: 5.6–14.7), of whom 40 (57.1%) received concurrent QS + ICI and 30 (42.9%) received QS alone. Table 1 describes patient characteristics.

Median age at the time of RT was 65.5 years (IQR: 57.9–77.8). Most common primary tumor sites were oropharynx (32.9%), oral cavity (24.3%), and larynx (15.7%). Age, race, sex, primary site, P16 status, smoking history, and performance status were well balanced between the two groups. T- and M-stage were statistically similar between the two groups, while the concurrent QS + ICI group had a significantly higher percentage of N2/3-stage patients (80.0% vs. 46.7%, *p* = 0.019). Twenty-four (34.3%) patients had undergone prior surgery on the primary site, while thirty-five (50.0%) patients had received prior radiation to a median dose of 66.0 Gy (IQR: 60.0–70.0). Patients received a median of three QS cycles (IQR: 3–4) with a median total RT dose of 44.4 Gy (IQR: 44.4–59.2). Median duration between QS cycles was 21 days (IQR: 21–21).

Of the 40 patients who received concurrent ICIs, 28 (70.0%) received Pembrolizumab, 11 (27.5%) received Nivolumab, and 1 (2.5%) received Cemiplimab. PD-L1 status was available for 21 patients, of which 15 (71.4%) had a combined positive score (CPS) ≥ 1%. Median CPS was 17.5 (range: 3–100) in patients with CPS ≥ 1. The median number of ICI cycles was eight (IQR: 4–12) in the QS + ICI group. Nineteen patients (27.1%) also received concurrent systemic therapy other than ICIs, which was significantly higher in patients who did not receive concurrent ICIs (50.0% vs. 10.0%, *p* < 0.001).

### 3.1. Patterns of Treatment Failure

Response rates and patterns of treatment failure are provided in detail in Table 2.

Thirty-four patients experienced progression, including fifteen with local progression. Local control rates for all patients at 12 and 24 months were 75.5% and 60.1%, respectively. Local control was significantly higher in the QS + ICI group (12-month: 84.7% vs. 63.3%, *p* = 0.038, Figure 2A). Distant control at 12 months was similar between the two groups (QS + ICI vs. QS alone: 56.4% vs. 63.2%, *p* = 0.629, Figure 2B). Median OS was 9.4 months and was similar between the two groups (QS + ICI vs. QS alone: 9.0 vs. 10.0 months, *p* = 0.850, Figure 2C). Overall, 71.4% patients had a complete or partial response, which was similar between the two groups (QS + ICI vs. QS alone: 70.0% vs. 73.3%, *p* = 0.487).

### 3.2. Clinical Factors Affecting Control and Survival

Table 3 describes the UVA and MVA Cox proportional hazard risks of clinical factors affecting local control.

In UVA, patients who had received prior systemic therapy had a significantly higher risk of local recurrence (HR, 10.073; 95% CI, 1.323–76.722; *p* = 0.026), while local recurrence was lower in patients who received concurrent ICIs with QS (HR, 0.337; 95% CI, 0.115–0.989; *p* = 0.048). In MVA, only concurrent QS + ICIs persisted as a significant predictor of local control (HR, 0.238; 95% CI, 0.073–0.778; *p* = 0.018).

As shown in Appendix A, no variable was significantly associated with distant control in UVA and MVA. As shown in Appendix A, primary site, smoking history, and Eastern Cooperative Oncology Group (ECOG) performance status ≥ 2 were each significantly associated with OS in UVA, while only ECOG performance status remained significant in MVA (HR, 1.844; 95% CI, 1.020–3.334; *p* = 0.018).

### 3.3. Adverse Events

No patient experienced CTCAE grade ≥ 4 toxicity. The highest graded toxicities were 1, 2, and 3 for 35.7%, 37.1%, and 22.9% of patients, respectively (see Table 4).

Grade 3 toxicity was similar between the two groups (QS + ICI vs. QS alone: 17.5% vs. 30.0%, *p* = 0.622). The most common acute grade ≥ 1 toxicity seen was dermatitis (34.3%), while the most common late grade ≥ 1 toxicity was xerostomia (52.8%). The most common acute grade 3 toxicity was dysphagia (32.9%), and the most common late grade 3 toxicities were radionecrosis (8.6%) followed by xerostomia (7.1%). All patients who developed radionecrosis (five with osteoradionecrosis, one with cerebral radionecrosis) had undergone prior local RT with overlapping radiation volumes. Rates of grade 3 toxicity were statistically similar between patients who underwent re-irradiation and those without previous RT (28.6% vs. 17.1%, *p* = 0.250). Only one patient had an immune-related adverse event (IRAE) in the concurrent QS + ICI group, which occurred 13 months after RT while on maintenance Pembrolizumab.

## 4. Discussion

This represents the first study to our knowledge evaluating the benefit of adding immunotherapy to QS radiation for patients with HNC. We observed a statistically significant improvement in local control in patients who received concurrent QS + ICIs, with a >20% increase in local control at 12 months. This local control benefit remained significant in both univariate and multivariable analyses.

Patients with advanced HNC often present with substantial tumor burden, resulting in significant symptoms. Local control is a key factor to consider in this setting, given the significant impact it can have on quality of life for these patients, both in alleviating presenting symptoms and preventing progression, which can lead to further morbidity. We observed an improvement in local control with concurrent QS + ICIs, with no increase in grade ≥ 3 toxicity. Paris et al., who published results from the first prospective trial of QS radiation for patients with HNC ineligible for curative treatment, reported a local control rate of 63% [18]. The 63% 1-year local control rate in our cohort of patients who received QS alone was identical to this study, while local control rates in our concurrent QS + ICI cohort were significantly higher at 85% and 75% at 1 and 2 years, respectively.

Prior prospective trials and retrospective analyses evaluating QS radiation for HNC have reported median OS rates ranging from 4.5 to 5.7 months [16,18,21]. The median OS in our cohort of 9.4 months compares favorably to these historical controls. This may reflect improvement in radiation techniques as well as systemic therapy options, as none of the prior studies included patients treated with immunotherapy, which has been shown to provide survival benefit in recurrent and metastatic settings for HNC [26]. In our study, we did not see an OS benefit in patients who received concurrent QS + ICI. This may be in part because of the limited sample size, and in part due to baseline differences between the two groups. Our patient population was also heavily pre-treated, with 60% patients having received prior systemic therapy. While the survival rate in our cohort was slightly lower than the median OS of 11.5 months shown in KEYNOTE-048, this may also be explained by the fact that our patients were heavily pre-treated and did not represent an identical patient cohort to that of the trial.

A recent NCDB analysis reported that 7% of patients with HNC treated with palliative RT receive QS [36]. Pre-clinical studies and early clinical studies suggest that a combination of immunotherapy with QS radiation may be more effective than either treatment alone in treating locally advanced HNC [37,38]. Results of the Phase 1b KEYNOTE-012 trial evaluating Pembrolizumab for recurrent and metastatic HNC reported an objective response rate of 18% [27]. Several other recent trials evaluating the addition of ICIs to chemoradiation for locally advanced HNC have shown limited improvement in survival outcomes, while the NRG-HN004 study showed a significantly worse locoregional failure rate [30,31,32]. These results suggest that elective nodal irradiation may create regional immunosuppression, thus blunting the immunostimulatory effects of ICIs, and thereby decreasing its efficacy. In our study, QS radiation targeted only gross tumor without elective nodal radiation, thus limiting the volume treated with radiation, which may potentially decrease the risk of immunosuppression. The objective response rate of 71% in our cohort compares favorably to the <20% response rates reported in trials for recurrent and metastatic HNC treated with systemic therapy alone [24,25,26,27,28,29].

In our study, administration of prior systemic therapy and concurrent QS + ICIs were both significant predictors of local control on UVA. Prior systemic therapy was detrimental to local control, likely secondary to the fact that patients previously treated with systemic therapy had failed prior lines of treatment and harbored more resistant tumor cell clones at the time of radiation. On multivariable modeling, only concurrent QS + ICI persisted as a significant predictor of local control with an HR of 0.238, implying a >75% relative risk reduction for local progression with the use of concurrent ICIs with QS.

This study is limited by its retrospective design, and therefore inherent biases that affect all retrospective studies, such as the selection bias of who received ICIs concurrent with QS. Similarly, due to the study’s retrospective nature, the ability to assess treatment-related toxicity in a systematic and comprehensive fashion is limited in comparisons between the two treatment arms. Another limitation is the sample size. To maintain a more homogenous patient population, we excluded patients who did not complete at least three cycles of QS radiation and those without post-treatment clinical follow-up or imaging. Accordingly, results from our study cannot be extrapolated to patients who receive <three cycles of QS radiation. Given its retrospective nature and the fact that we did not perform matched cohort comparisons, definitive conclusions cannot be made on the superiority of the addition of ICIs to QS. While stratification of baseline characteristics using a 2 × 2 analysis of patients who had received prior RT and prior ICIs would potentially strengthen conclusions that may be drawn from the data, the relatively small number of patients that would be included in each cohort would limit the power of statistical analysis. We did attempt to address this by comparing baseline variables between those who did and did not receive concurrent ICIs, showing no significant differences, such as a similar percentage of patients who had received previous radiation and prior systemic therapy. Given that concurrent ICIs significantly improved local control despite having a significantly lower percentage of patients receiving other concurrent systemic therapy (chemotherapy and/or cetuximab), with the two groups otherwise well balanced, our data suggest a potential added benefit of concurrent ICIs in combination with QS radiation. All patients were treated at a single institution, and analysis from additional cohorts providing corroborating evidence would strengthen conclusions. Further prospective randomized studies to validate our findings are warranted. Two ongoing phase II trials (NCT04454489, NCT04373642) are evaluating the efficacy and tolerability of combination ‘Quad Shot’ palliative radiotherapy with Pembrolizumab immunotherapy for advanced, recurrent, and/or metastatic head and neck cancer [34,39].

## 5. Conclusions

The combination of QS with concurrent ICIs was well tolerated and significantly improved local control compared to QS alone. The median OS of 9.4 months compares favorably to historical controls from other studies reporting outcomes of QS RT. This approach represents a promising treatment option for patients with HNC who are not candidates for curative-intent treatment and warrants further prospective evaluation.

## Figures and Tables

**Figure 1 cancers-16-01049-f001:**
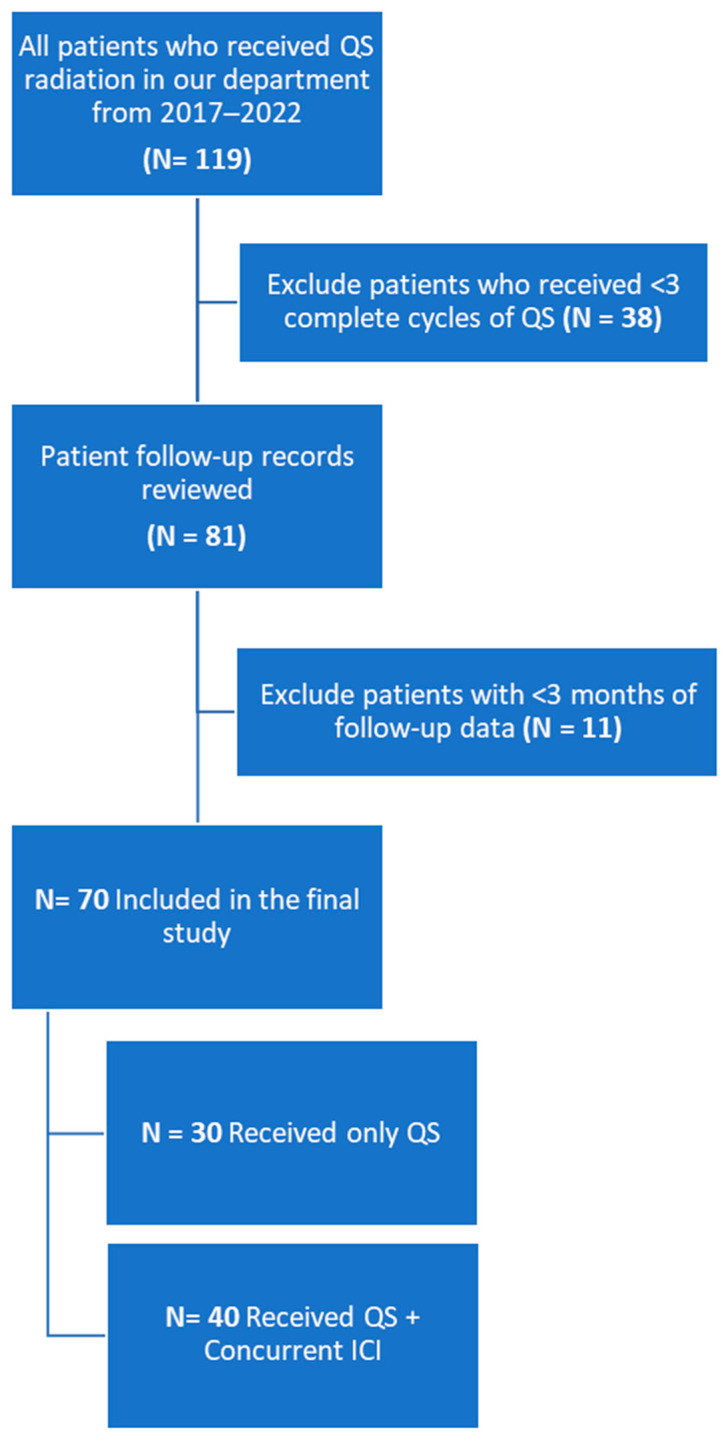
Patient selection.

**Figure 2 cancers-16-01049-f002:**
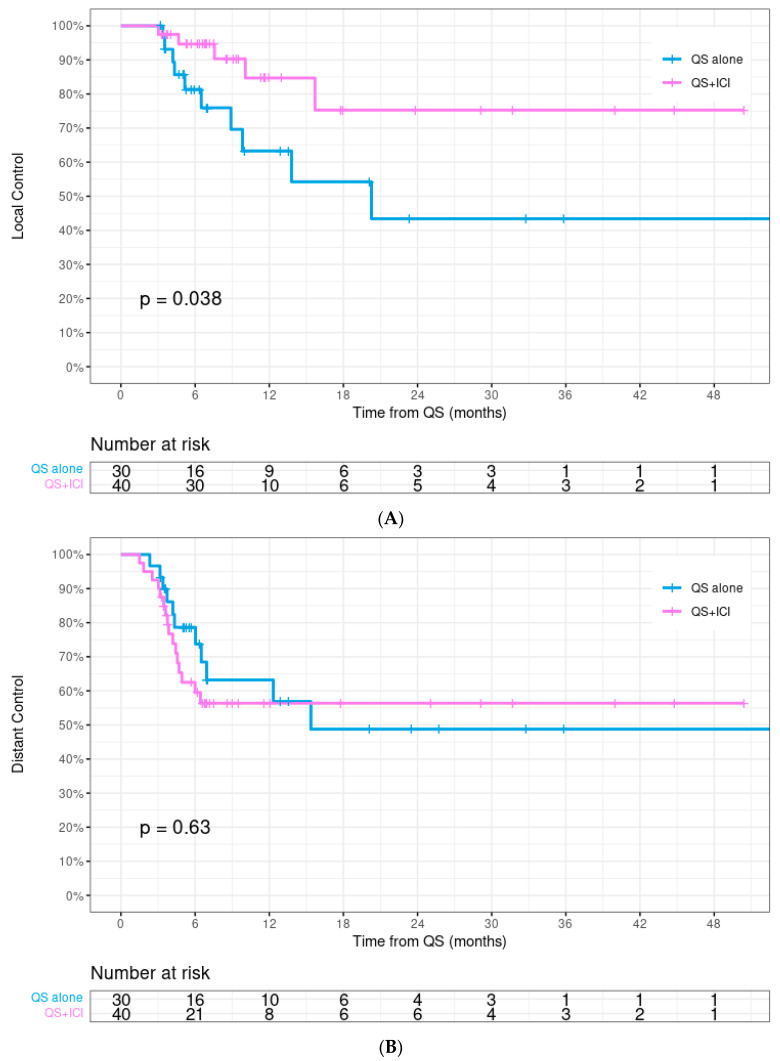
(**A**–**C**) Kaplan–Meier curves comparing ‘Quad Shot’ radiation alone (QS alone) to QS with concurrent immune checkpoint inhibitors (QS + ICI) showed (**A**) local control was significantly higher with concurrent QS + ICI (12-month: 85% vs. 63%, *p* = 0.038). (**B**) Distant control and (**C**) overall survival were similar between the two groups.

**Table 1 cancers-16-01049-t001:** Patient characteristics.

Characteristic	All Patients (n = 70) n (%)	QS + ICI (n = 40) n (%)	QS Alone (n = 30) n (%)	*p*-Value
Age (y): median (IQR)	65.5 (57.9–77.8)	63.3 (57.3–70.8)	67.2 (59.9–80.7)	0.410
Race				0.219
White	60 (85.7)	36 (90.0)	24 (80.0)	
African American	8 (11.4)	4 (10.0)	4 (13.3)	
Others	2 (2.9)	0	2 (6.7)	
Sex				0.366
Male	56 (80.0)	30 (75.0)	26 (86.7)	
Female	14 (20.0)	10 (25.0)	4 (13.3)	
Primary site				0.371
Oropharynx	23 (32.9)	12 (30.0)	11 (36.7)	
Oral cavity	17 (24.3)	8 (20.0)	9 (30.0)	
Larynx	11 (15.7)	8 (20.0)	3 (10.0)	
Cutaneous	7 (10.0)	4 (10.0)	3 (10.0)	
Hypopharynx	4 (5.7)	1 (2.5)	3 (10.0)	
Paranasal sinus	3 (4.3)	3 (7.5)	0	
Thyroid	3 (4.3)	2 (5.0)	1 (3.3)	
Major salivary glands	2 (2.9)	2 (5.0)	0	
P16-positive oropharynx	11 (55.0)	7 (58.3)	4 (50.0)	0.462
PD-L1 CPS ≥ 1%	15/21 (71.4)	10/13 (76.9)	5/8 (62.5)	0.683
T-stage				0.164
T0/Tx	6 (8.6)	3 (7.5)	3 (10.0)	
T1–2	9 (12.9)	4 (10.0)	5 (16.7)	
T3–4	55 (78.5)	33 (82.5)	22 (73.3)	
N-stage				0.019 *
N0–1	24 (34.3)	8 (20.0)	16 (53.3)	
N2–3	46 (65.7)	32 (80.0)	14 (46.7)	
M-stage				0.415
M1	18 (25.7)	12 (30.0)	6 (20.0)	
Smoking history				0.384
Current	15 (21.4)	8 (20.0)	7 (23.3)	
Former	41 (58.6)	26 (65.0)	15 (50.0)	
None	14 (20.0)	6 (15.0)	8 (26.7)	
Smoking pack years: median (IQR)	30 (18–50)	25 (18–50)	44 (18–50)	0.487
ECOG performance status				0.316
0	10 (14.3)	6 (15.0)	4 (13.3)	
1	35 (50.0)	22 (55.0)	13 (43.3)	
2	19 (27.1)	12 (30.0)	7 (23.3)	
3	6 (8.6)	0	6 (20.0)	
Surgery for primary site	24 (34.3)	17 (42.5)	7 (23.3)	0.077
Prior systemic therapy	42 (60.0)	25 (62.5)	17 (56.7)	0.402
ICI	13 (31.0)	5 (20.0)	8 (47.1)	
Chemotherapy alone	19 (45.2)	14 (56.0)	5 (29.4)	
Cetuximab alone	1 (2.4)	0	1 (5.9)	
Chemotherapy + Cetuximab	9 (21.4)	6 (24.0)	3 (17.6)	
Prior radiation therapy	35 (50.0)	22 (55.0)	13 (43.3)	0.235
Concurrent non-ICI systemic therapy	19 (27.1)	4 (10.0)	15 (50.0)	<0.001 *
Chemotherapy alone	9 (12.9)	3 (7.5)	6 (40.0)	
Cetuximab alone	4 (5.7)	0	4 (26.7)	
Chemotherapy + Cetuximab	6 (8.6)	1 (2.5)	5 (33.3)	
Number of QS cycles				0.507
3	52 (74.3)	30 (75.0)	22 (73.3)	
4	17 (24.3)	10 (25.0)	7 (23.3)	
5	1 (1.4)	0	1 (3.3)	

Abbreviations: QS = ‘Quad Shot’; ICI = immune checkpoint inhibitor; IQR = interquartile range; ECOG = Eastern Cooperative Oncology Group. * Considered statistically significant based on *p*-value < 0.050.

**Table 2 cancers-16-01049-t002:** Patterns of failure and survival outcomes.

Outcome	All Patients (n = 70) n (%)	QS + ICI (n = 40) n (%)	QS Alone (n = 30) n (%)	*p*-Value
Objective response				0.487
CR	16 (22.8%)	8 (20.0%)	8 (26.7%)	
PR	34 (48.6%)	20 (50.0%)	14 (46.7%)	
SD	13 (18.6%)	8 (20.0%)	5 (16.7%)	
PD	7 (10.0%)	4 (10.0%)	3 (10.0%)	
Local control				0.038 *
12-month	75.5%	84.7%	63.3%	
24-month	60.1%	75.3%	43.4%	
Distant control				0.629
12-month	59.4%	56.4%	63.2%	
24-month	51.9%	56.4%	48.8%	
Overall survival				0.850
12-month	35.8%	30.0%	43.6%	
24-month	23.2%	21.8%	20.3%	
Median (95% CI)	9.4 m (6.5–12.2)	9.0 m (6.7–11.4)	10.0 m (5.5–14.5)	

Abbreviations: QS = ‘Quad Shot’; ICI = immune checkpoint inhibitor; CR = complete response; PR = partial response; SD = stable disease; PD = progressive disease; CI = confidence intervals; m = months. * Considered statistically significant based on *p*-value < 0.050.

**Table 3 cancers-16-01049-t003:** Univariate and multivariable analyses of factors affecting local control.

	Univariate Analysis		Multivariable Analysis	
	HR (95% CI)	*p*-Value	HR (95% CI)	*p*-Value
Age	1.014 (0.003–319.381)	0.996	Not included	
Race	0.594 (0.166–2.125)	0.423	Not included	
Sex	1.418 (0.451–4.460)	0.550	Not included	
Primary site	0.911 (0.773–1.074)	0.267	Not included	
T-stage	0.618 (0.171–2.236)	0.464	Not included	
N-stage	3.059 (0.683–13.699)	0.144	1.729 (0.905–3.305)	0.097
M-stage	0.334 (0.074–1.499)	0.152	0.397 (0.086–1.838)	0.237
Smoking	1.899 (0.523–6.897)	0.330	Not included	
ECOG PS ≥ 2	1.051 (0.325–3.403)	0.934	Not included	
Surgery	1.130 (0.385–3.312)	0.824	Not included	
Prior ST	10.073 (1.323–76.722)	0.026 *	7.035 (0.917–54.001)	0.061
Conc QS + ICI	0.337 (0.115–0.989)	0.048 *	0.238 (0.073–0.778)	0.018 *
Prior RT	1.100 (0.398–3.040)	0.854	Not included	
Adjuvant ICI	0.978 (0.332–2.877)	0.968	Not included	
No. of QS cycles	9507.71 (0.000–5.2 × 10^151^)	0.958	Not included	

Abbreviations: HR = hazard ratio; CI = confidence intervals; ECOG PS = Eastern Cooperative Oncology Group performance status; ST = systemic therapy; conc = concurrent; QS = ‘Quad Shot’; ICI = immune checkpoint inhibitor; RT = radiation therapy. * Considered statistically significant based on *p*-value < 0.050.

**Table 4 cancers-16-01049-t004:** Toxicity profile.

Toxicity	Grade 1 n (%)	Grade 2 n (%)	Grade 3 n (%)	Any Grade n (%)
Acute				
Dermatitis	18 (25.7)	5 (7.1)	1 (1.4)	24 (34.3)
Mucositis	10 (14.3)	8 (11.4)	1 (1.4)	19 (27.1)
Dysphagia	6 (8.6)	12 (17.1)	5 (7.1)	23 (32.9)
Dysgeusia	10 (14.3)	4 (5.7)	1 (1.4)	15 (21.4)
Nausea/Vomiting	1 (1.4)	0	0	1 (1.4)
Pain	1 (1.4)	2 (2.9)	1 (1.4)	4 (5.7)
Late				
Xerostomia	25 (35.7)	11 (15.7)	1 (1.4)	37 (52.8)
Radionecrosis	0	0	6 (8.6)	6 (8.6)
Lymphedema/Fibrosis	1 (1.4)	2 (2.9)	0	3 (4.3)

## Data Availability

Research data are stored in an institutional repository and will be shared upon request to the corresponding author.

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
