# Peer review of "Palliative Quad Shot Radiation Therapy with or without Concurrent Immune Checkpoint Inhibition for Head and Neck Cancer"

_cancers, 2024, doi:10.3390/cancers16051049_

Round 1

Reviewer 1 Report

Comments and Suggestions for Authors

The authors present a retrospective analysis of 70 patients with recurrent/metastatic HNC or those ineligible for curative treatment who were treated with quad shot (QS) hypofractionated palliative RT with or without immune checkpoint inhibitors (ICI). Although this is a retrospective study, it is well designed and clearly presented. The groups are comparable in terms of the most important prognostic factors. The results show that local control is significantly better in the group that received ICI in addition to QS-RT, while there were no differences between the two groups in terms of distant control or overall survival. The discussion is structured and comprehensively summarizes the study results in the context of existing knowledge and critically addresses their limitations. The conclusions are appropriate and based on the results.

I have the following comments:

1.       Another limitation is certainly the retrospective assessment of treatment toxicity, which as such cannot be systematic and comprehensive.

2.       The authors cited a follow-up period (FUP) of < 3 months as an exclusion criterion: they should have included all patients with completed FUP, including those who died within 3 months of FUP. Early deaths are an indicator of excessive toxicity/inefficacy of therapy.

3.       Another exclusion criterion was < 3 cycles of QS-RT: in this way, patients in whom the QS regimen was excessively toxic or ineffective were excluded, leading to an additional bias in the analysis.

Reviewer 2 Report

Comments and Suggestions for Authors

The authors retrospectively analyzed their QS-treated HNC patients with or without ICI. They demonstrated that QS with ICI improved local control, but not OS. Through MVA, ICI was significantly associated with better local control. Their observation unveiled the possibility of QS with ICI as a promising treatment option for recurrent and metastatic HNC. Their analyses are well designed and refined, giving an impact for head and neck oncologists. Only one minor issue should be checked.

Line 69, 14.4Gy should be 14.8Gy.

Round 2

Reviewer 1 Report

Comments and Suggestions for Authors

The authors responded appropriately to the criticisms/comments and supplemented the manuscript. I have no further questions/comments.